# Integrated Transcriptome and Metabolome Dissecting Interaction between *Vitis vinifera* L. and Grapevine Fabavirus

**DOI:** 10.3390/ijms24043247

**Published:** 2023-02-07

**Authors:** Baodong Zhang, Mengyan Zhang, Xiaojun Jia, Guojun Hu, Fang Ren, Xudong Fan, Yafeng Dong

**Affiliations:** National Center for Eliminating Viruses from Deciduous Fruit Trees, Research Institute of Pomology, Chinese Academy of Agriculture Sciences, Xingcheng 125100, China

**Keywords:** GFabV concentration, plant defense, metabolome, transcriptome, grapevine

## Abstract

Grapevine fabavirus (GFabV) is a novel member of the *Fabavirus* genus associated with chlorotic mottling and deformation symptoms in grapevines. To gain insights into the interaction between GFabV and grapevines, *V. vinifera* cv. ‘Summer Black’ infected with GFabV was investigated under field conditions through physiological, agronomic, and multi-omics approaches. GFabV induced significant symptoms on ‘Summer Black’, and caused a moderate decrease in physiological efficiency. In GFabV-infected plants, alterations in carbohydrate- and photosynthesis-related genes might trigger some defense responses. In addition, secondary metabolism involved in plant defense was progressively induced by GFabV. Jasmonic acid and ethylene signaling were down-regulated in GFabV-infected leaves and berries along with the expression of proteins related to LRR and protein kinases, suggesting that GFabV can block the defense in healthy leaves and berries. Furthermore, this study provided biomarkers for early monitoring of GFabV infection in grapevines, and contributed to a better understanding of the complex grapevine-virus interaction.

## 1. Introduction

Grapevine is one of the most widely cultivated economic fruit crops, whose global production reached nearly 14.8 million tons in 2020 (FAO). Grapes, either table or wine grapes, have great health beneficial effects due to their abundant anticarcinogenic polyphenols such as resveratrol [1]. However, grapevines are easily infected by viruses, and more than 80 grapevine-infecting viruses have been reported in the world [2]. Some newly discovered viruses, such as grapevine pinot gris virus (GPGV) [3], grapevine red blotch virus (GRBV) [4], grapevine vein clearing virus (GVCV) [5], and grapevine fabavirus (GFabV) [6], can cause serious symptoms. Among these new viruses, GRBV has been extensively studied for its effects on physiology, berry, and wine composition [4], while the effects of other viruses on grape physiology and quality have not been well documented.

Grapevine fabavirus (GFabV) is a new member of the *Fabavirus* genus in the *Secoviridae* family, which was first identified in ‘Black Beet’ (BB) and ‘Nagano Purple’ (NP) by high-throughput next-generation sequencing (NGS) [6]. To date, GFabV has been reported in the United States, Japan, China, and South Korea [6,7,8]. Fan et al. for the first time discovered GFabV on native grapes in China, and preliminarily suggested that GFabV is associated with the chlorotic mottling and deformation symptoms of ‘Beta’ grapevines [7]. Subsequent studies have suggested that GFabV may be associated with the outbreak of ‘Shine Muscat’ (*V. vinifera* cv. ‘Akitsu-21’ × *V. vinifera* cv. ‘Hakunan’) virus disease in China [9]. The original species of Shine Muscat grapevine cultivated in Japan does not carry the virus, but the female parent tree grafted on the 5BB rootstock does, leading to the prevalence of ‘Shine Muscat’ virus disease [10]. Furthermore, the Chinese isolates of GFabV XF, CXZ, and BETA2 have also been successively reported [11,12].

With the development of high-throughput sequencing technology, transcriptomes have been widely used to analyze plant-virus interactions. Some researchers have explored the effect of viruses on grapevines by RNA sequencing. For example, a previous study elucidated the global gene expression patterns during fruit development in GLRaV-3 infected and uninfected grapevines, as well as demonstrated the molecular mechanisms related to plant response to the virus [13]. Recently, some new insights into the infection mechanism of leafroll were obtained by using physiological and molecular approaches [14]. The molecular interaction between grapevine rupestris stem pitting-associated virus (GRSPaV) and grapevines was illustrated by ecophysiological and transcriptome methods, and some overlapping differentially expressed genes were found between GRSPaV infection and abiotic stresses such as drought and salinity [15]. In addition, *V. vinifera* cv. Albarossa infected with grapevine virus B (GVB) was analyzed under field conditions through transcriptome and sugar metabolome analysis combined with agronomic study. The results suggested that in GVB-infected plants, the accumulation of soluble carbohydrates in the leaves and transcriptional changes in sugar- and photosynthesis-related genes seemed to trigger defense responses [16]. Furthermore, GFLV induced a strain- and variety-specific defense response similar to hypersensitivity as indicated by transcriptomic and metabolomic profiling alone [17].

This study aims to clarify the interaction between GFabV and *V. vinifera* plants. To this end, GFabV-susceptible *V. vinifera* cv. ‘Summer Black’ was chosen as the object for comparison of the differences between GFabV-infected and GFabV-free plants through transcriptome and widely-targeted metabolome analysis combined with physiological and agronomic approaches. The findings may contribute to a better understanding of host-virus interactions.

## 2. Results

### 2.1. GFabV Concentration in Infected Grapevines

In 2019, the concentration of GFabV RNA at different growth stages was measured in GFabV-infected ‘Summer Black’ grapevines by quantitative RT–PCR. As a result, there were wide variations in GFabV concentration in infected leaves among different stages in 2019, with the lowest level of viral RNA being detected in July 2019 (EL31), which was about 10-fold lower than the value observed in the leaves of the first developmental stage (EL15) (Figure 1). Interestingly, the viral RNA content significantly increased at the fifth and sixth developmental stages (EL35, EL38; 2019).

### 2.2. Physiological and Agronomic Performance of GFabV-Infected Grapevines

In 2019, GFabV-infected plants had significantly lower chlorophyll content and net photosynthesis (P_n_) than GFabV-free plants at different stages (Figure 2a,b). C_i_ was similar in the two stages and was related to a decrease in P_n_ (Figure 2c). As expected, g_s_ was consistently lower in GFabV-infected plants (Figure 2d). The transpiration (E) of GFabV-infected plants decreased at the EL15 stage, while it increased at the EL31 stage relative to that of GFabV-free plants (Figure 2e).

In terms of agronomic traits and fruit quality indicators, the results were generally consistent between the two experimental years. In particular, GFabV-infected and GFabV-free plants showed significant differences in berry length and total cyanine glycosides, with no obvious difference in bunch length (Table 1).

### 2.3. Changes in the Aroma Compounds between GFabV-Infected and GFabV-Free Grapevines

Aroma compounds in grapes are secondary metabolites with great importance for grapevine quality. To gain insights into the difference in aroma between GFabV-infected and GFabV-free plants at EL38, GC–MS was performed to identify aroma compounds. A total of 35 and 39 aroma compounds were identified in GFabV-free and GFabV-infected plants, respectively. Alcohol was the dominant aroma compound in GFabV-infected plants. Interestingly, some esters were only detected in GFabV-free plants, while some terpenes were exclusively found in GFabV-infected plants. In addition, significant differences in phenylethyl alcohol and cis-Ocimenol were found between GFabV-free and GFabV-infected plants, while 5-hydroxymethylfurfural was only identified in GFabV-infected grapevines (Appendix A).

### 2.4. RNA–Seq Analysis

GFabV-infected and GFabV-free leaf samples were collected at two developmental stages (EL15 and EL31), and berry samples were collected at EL38 in 2019. The Illumina novaseq 6000 platform was used to sequence the transcriptome of different leaf and berry samples at different stages.

In the leaves, twelve RNA–seq data sets were generated, yielding a total of 73.42 Giga base (Gb) clean reads, with a maximum of 5.88 Gb for each sample, and the Q30 of clean reads was above 93.86% (Appendix A). Compared with the reference genome (International Grapes Genomes Program, IGGP, www.vitaceae.org, accessed on 12 December 2020, 12x.45version), 1492 new transcripts were identified, among which 1167 were annotated. To predict the functions of all genes, we blasted the assembled sequences using various databases, including COG, GO, KEGG, KOG, Pfam, Swiss-prot, eggNOG, and NR, generating 30113 functionally annotated genes (Appendix A). 

For the transcripts in berries, six RNA–seq data set were obtained. A total of 38.15 Gb clean reads were generated, and each sample had an average of 5.73 Gb of clean reads with a Q30 score of 93.19% or above (Appendix A). By comparison with the above reference genome, we identified 1248 new transcripts, among which 969 were annotated. To understand the functions of all genes, we searched for the above databases, and as a result, 29915 unigenes were annotated (Appendix A).

The differentially expressed genes (DEGs) were grouped into 11 functional classes based on the biological process they are involved in by blasting the GO classification. GFabV-infected grapevines showed significant differences in the regulation of transcripts between leaves and berries at two developmental stages (Figure 3).

#### 2.4.1. Global Gene Expression Changes in GFabV-Infected Leaves

DEGs were identified with the following thresholds: log_2_|fold change| > 1.5 and *p* value < 0.05. There were 514 DEGs (116 up-regulated and 398 down-regulated) at the EL15 stage and 250 DEGs (141 up-regulated and 109 down-regulated) at the EL31 stage between GFabV-infected and GFabV-free plants (Appendix A). Interestingly, 16 DEGs were commonly identified between EL15 down-regulated and EL31 up-regulated genes, mainly including aldehyde dehydrogenase family 2 (VIT_01s0026g00220), LRR receptor-like serine/threonine-protein kinase (VIT_00s0125g00090), and 1-aminocyclopropane-1-carboxylate oxidase (ACO, VIT_00s2086g00010) (Appendix A). In addition, one DEG was commonly detected between EL15 down-regulated and EL31 down-regulated genes, which was not annotated by searching in the above several databases (Appendix A). 

Notably, several up-regulated genes involved in secondary metabolism at the EL15 stage were annotated as phenylpropanoid (ko00940), flavone and flavonol biosynthesis (ko00944), flavonoid (ko00941), and diterpenoid (ko00904) using the Kyoto Encyclopedia of Genes and Genomes (KEGG) database (Figure 4a). These genes were involved in defense activation. Only a few up-regulated genes implicated in disease and stress response were identified at the EL15 stage, such as heat shock protein 83 (HSP, VIT_02s0025g00280) and cationic peroxidase 1 (VIT_06s0004g07750). However, one gene (phenylalanine ammonia-lyase, PAL, VIT_11s0016g01520) was observed to be down-regulated. In addition, GFabV infection down-regulated many genes involved in defense response, including the LRR superfamily, NAC domain-containing proteins, and glutathione S-transferas. Further statistical tests showed that the genes involved in ethylene-, auxin-, and gibberellin-mediated signaling pathways at the EL15 stage were down-regulated, such as VIT_00s2086g00010 (Table 2; Appendix A).

The genes involved in ‘transport’ were up-regulated at the EL31 stage. It was observed that the transcription factors of WRKY, MYB, and AP2/ERF families were positively regulated, and a large number of genes involved in signal transduction and hormones were up-regulated at the EL31 stage. Furthermore, several genes involved in cell wall organization were up-regulated, such as cellulose synthase-like protein G3 (VIT_02s0025g01850). On the contrary, some genes involved in carbohydrate metabolism were down-regulated at EL31, such as glucose-1-phosphate adenylyltransferase large subunit 1 (VIT_03s0038g04570). However, a gene encoding photosystem II protein D1 (VIT_13s0019g02630) was up-regulated, and several transcripts related to carbon fixation in photosynthesis were down-regulated. We observed that those genes down-regulated at the EL31 stage of infected leaves were involved in lipid and secondary metabolism. Interestingly, a gene encoding tubulin beta-6 chain (VIT_06s0009g02020) involved in microtubules was down-regulated, suggesting that the cell cycle was responsive to GFabV (Table 2; Appendix A).

#### 2.4.2. Global Gene Expression Changes in GFabV-Infected Berries

In berries, we identified 238 DEGs (128 up-regulated, 110 down-regulated) between GFabV-infected and GFabV-free plants at the EL38 stage using the above parameters (Appendix A).

Most up-regulated genes were related to lipid, hormones (auxin-and gibberellin-mediated signal transduction), amino acid metabolism, and the cell cycle. In addition, some transcripts rich in phenylpropanoid (ko00940), flavone and flavonol biosynthesis (ko00944), and flavonoid (ko00941) were up-regulated, though a gene encoding flavonol synthase/flavanone 3-hydroxylase (VIT_18s0001g03470) was repressed (Figure 5a; Table 2). The genes involved in transcription factors (PHL, bHLH) were up-regulated. In contrast, in the photosynthesis metabolism, some genes related to photosystem II proteins were down-regulated, indicating the inhibition of photosynthesis. Most genes related to defense and stress response were negatively regulated (such as heat shock protein and disease resistance protein), while a gene encoding peroxidase15 (VIT_06s0004g01180) was up-regulated. Furthermore, several down-regulated genes were identified in the carbohydrate metabolism. For instance, a gene encoding UDP-glycosyltransferase 92A1 (VIT_05s0062g01280), a member of the UDP glycosyltransferases (UGTs) family, catalyzes lipophilic compounds to produce glycosides (Table 2; Appendix A).

### 2.5. Metabolomic Analysis

To summarize the data, heatmaps were generated for GFabV-infected and GFabV-free leaves (at the EL15, EL31 stage) and berries. In leaves, 595 metabolites were detected by the UPLC–MS/MS (Appendix A). These metabolites from each sample were identified through a hierarchical clustering heatmap, which could highlight all metabolites differentially produced in the two periods (EL15 and EL31) (Figure 6a). In particular, most metabolites at the EL31 stage were down-regulated in GFabV-infected grapevine. Furthermore, the Pearson correlations among the biological replicates of leaves were 0.91–1 at EL15, while 0.92–1 for GFabV-free leaves and 0.81–1 for GFabV-infected leaves at EL31 (Figure 6b).

In berries, we detected 659 metabolites by UPLC–MS/MS (Appendix A). For a series of pairwise comparisons of metabolite expression, a cluster analysis of differential metabolite expression was performed. The hierarchical clustering results revealed that the expression of most metabolites was different between GFabV-infected and GFabV-free plants (Figure 7a). Additionally, the Pearson correlation coefficients of these metabolites were 0.8–1 between GFabV-infected and GFabV-free grapevines (Figure 7b).

Differentially expressed metabolites (DEMs) were detected using the following thresholds: |fold change| > 1, *p* value < 0.05, and variable importance in the prediction (VIP) ≥ 1. As a result, we screened 27 DEMs (14 up-regulated, 13 down-regulated) at the EL15 stage and 46 DEMs (10 up-regulated, 36 down-regulate) at the EL31 stage in GFabV-infected leaves (Appendix A). In addition, we identified 229 DEMs in GFabV-free leaves and 238 DEMs in GFabV-infected leaves between the two stages. Furthermore, Venn diagrams were created between different stages or between GFabV-infected and GFabV-free plants (Appendix A). One common metabolite (3-phospho-D-glyceric acid) was identified between EL15-Down and EL31-Up, and another common metabolite (1-α-linolenoyl-glycerol-3-O-glucoside) was detected between EL15-Down and EL31-Down (Appendix A). A total of 127 common metabolites were identified between GFabV-infected and GFabV-free plants (Appendix A). The heatmap showed differential accumulation of metabolites (Figure 8a). Compared with those in GFabV-free leaves, most metabolites in GFabV-infected leaves were down-regulated, while shikimic acid at EL31 was up-regulated (Appendix A). Furthermore, a principal component analysis (PCA) was performed between EL15 and EL31. As a result, PCA1 could account for 49.7% of the total variance between stages (Appendix A).

The OPLS–DA score plot presents a good separation between GFabV-infected and GFabV-free leaves at the EL15 stage (Appendix A). A further analysis of DEMs at the EL15 stage demonstrated that these metabolites were mainly enriched in carbon fixation in photosynthetic organisms, starch and sucrose metabolism, glycolysis/gluconeogenesis, phenylalanine, tyrosine, and tryptophan biosynthesis (Figure 8b). In particular, phenylpyruvic acid (PPA), a precursor of phenyllactic acid (PLA), was down-regulated. A large number of amino acids, such as L-isoleucine and L-tartaric acid, were accumulated in infected leaves at the EL15 stage (Appendix A).

Similarly, the OPLS–DA score plot also shows a very good separation between GFabV-infected and GFabV-free leaves at the EL31 stage (Appendix A). Interestingly, organooxygen compounds decreased at EL15 but were accumulated at EL31. A comparative analysis of DEMs at EL31 was performed using KEGG pathways. DEMs were notably enriched in flavone and flavonol biosynthesis, glycolysis/gluconeogenesis, alpha-linolenic acid metabolism, phenylalanine metabolism, and plant hormone signal transduction (Figure 8c). Particularly, naringenin-7-O-glucoside (naringin) and myricetin (MYR), which are associated with the flavonoid pathway, were accumulated in infected leaves. Infected leaves showed decreases in cinnamic acids and their derivatives, including cinnamic acid, 2-hydroxycinnamic acid, and 3,4,5-trimethoxycinnamic acid, as well as in most carboxylic acids and their derivatives (Appendix A).

In berries, we screened 18 DEMs (8 up-regulated, 10 down-regulated) using the same threshold above (Appendix A). The OPLS–DA score plot represents statistically significant discrimination between GFabV-infected and GFabV-free berries (Appendix A). The heatmap also demonstrates similar results (Figure 9a). Furthermore, pathway profiling of DEMs was implemented using the KEGG database. As a result, these DEMs were mainly associated with flavone and flavonol biosynthesis, carbon fixation in photosynthetic organisms, and linolenic acid metabolism (Figure 9b). Most fatty acyl groups, including 9-oxononanoic acid and 13S-hydroperoxy-9Z, 11E-octadecadienoic acid, decreased in infected berries, while 12,13-epoxy-9-octadecenoic acid showed obvious accumulation in infected berries. In addition, 4-O-methylgallic acid, a major gallic acid metabolite, decreased in infected berries (Appendix A).

### 2.6. Combined Analysis of Transcriptome and Metabolome

To gain insights into the association between transcriptome and metabolome, we investigated the connection between DEGs and DEMs. At the EL15 stage, we constructed a co-expression network using a total of 514 DEGs and 31 DEMs with a |Pearson correlation| > 0.8 (Appendix A), which could provide some clues for the connection between DEMs and partially, DEGs. For instance, the up-regulated L-isoleucine metabolism was targeted by six DEGs, including VIT_17s0000g09800, VIT_04s0008g06930, Vitis_vinifera_newGene_1051, VIT_18s0001g08220, VIT_18s0166g00250, and VIT_11s0016g02190 (Appendix A).

The overall variations in genes and metabolites between GFabV-infected and GFabV-free leaves at EL15 are presented in Figure 10a. It could be observed that the DEGs and DEMs related to photosynthesis decreased in infected leaves. For example, 3-phospho-D-glyceric acid, which participates in carbon fixation in photosynthetic organisms, was decreased in infected leaves, and several down-regulated genes involved in carbon fixation in photosynthetic organisms were identified, such as the genes encoding phosphoenolpyruvate carboxykinase (VIT_00s2576g00010), glyceraldehyde-3-phosphate dehydrogenase 2, cytosolic (VIT_01s0010g02460), and mitochondrial aspartate aminotransferase (VIT_08s0058g01000). Similarly, changes in glycolysis were observed in GFabV-infected leaves. For example, a gene encoding mitochondrial aldehyde dehydrogenase family 2 member B4 (VIT_01s0026g00220) involved in glycolysis/gluconeogenesis was down-regulated. Moreover, a decrease in glycolysis-related metabolites was observed.

The most significant response at the transcriptional level was the defensive response to the invading virus. Notably, most R genes including LRR (leucine-rich repeats) were down-regulated under infection, such as leucine-rich repeat protein 2 (VIT_03s0091g00560). The expression of pathogenesis-related protein (VIT_14s0081g00030, VIT_10s0003g00580) and the plant receptor kinase (VIT_00s0398g00020) was decreased in infected leaves. Additionally, some genes related to oxidative stress response, including glutathione S-transferase (VIT_17s0000g02950) and chitinase (VIT_05s0094g00350), were also down-regulated in infected leaves. In response to GFabV, lignin is generally synthesized to strengthen the cell wall. Expression of the gene encoding cationic peroxidase 1 (VIT_06s0004g07750) was increased in infected leaves. Monolignol biosynthesis-related genes, such as cinnamoyl-coA reductase 1 (VIT_06s0004g02370), were up-regulated. These transcriptomic results suggested that GFabV infection triggers a defensive response, which could be confirmed by the metabolome data. For example, 1-α-Linolenoyl-glycerol-3-O-glucoside was down-regulated in infected leaves.

The same analysis was performed at the EL31 stage. We used cytoscape to build the network. A network with a |correlation coefficient| > 0.8 between 250 DEGs and 54 DEMs was established to dissect the relation between DEGs and DEMs. As shown in Appendix A, 46 DEMs and 66 DEGs formed a network. For these DEMs and DEGs, we further analyzed the changes in important biological pathways. In the Calvin cycle, we observed the metabolite NADP (nicotinamide adenine dinucleotide phosphate) was positively regulated, and the stomatal conductance at EL31 in GFabV-infected leaves was reduced, indicating inhibition of photosynthesis. Although some genes such as photosystem II protein D1 (VIT_13s0019g02630) and chloroplastic calvin cycle protein CP12-3 (VIT_18s0122g00460) showed increases in expression in infected leaves, the infection actually reduced plant photosynthesis. The glycolysis-related genes (such as VIT_08s0007g07600 and VIT_18s0001g15450) were negatively modulated by the infection. In addition, a large number of genes involved in defense were repressed by the viral infection, including the genes encoding l-ascorbate peroxidase, cytosolic (VIT_08s0040g03150), NAC domain-containing protein 71 (VIT_02s0012g01040), heat shock 70 kDa protein 8 (VIT_05s0020g03330), and LRR receptor-like serine/threonine-protein kinase at1g56130 (VIT_00s0125g00090).

Furthermore, we observed that a gene encoding cellulose synthase-like protein G3 (VIT_02s0025g01850) related to cell wall was up-regulated, which was accompanied by the positive regulation of lignin synthesis, suggesting that these metabolites are synthesized under viral infection. Starch metabolism was modified by the up-regulated genes, such as trehalose-phosphate phosphatase F (VIT_00s0233g00030), and Fructokinase-2 (VIT_05s0102g00710). We observed that shikimic acid was positively modulated by the HPLC-MS/MS, and lipid metabolism was up-regulated as well. Furthermore, some genes (such as VIT_04s0023g02900 and VIT_03s0017g02110) involved in the flavonoid pathway were up-regulated. Metabolome data showed the expression accumulation of naringenin-7-O-glucoside (Prunin), indicating that the flavonoid pathway was induced during virus infection. An increase in hormone metabolism was identified, including a gene encoding 1-aminocyclopropane-1-carboxylate oxidase (ACO, VIT_00s2086g00010) (Appendix A). 

To understand the relationship between DEGs and DEMs in berries, we used 238 DEGs and 18 DEMs with |correlation coefficient| > 0.8 to construct a network. As shown in Appendix A, 18 DEMs were linked to 52 DEGs. We further analyzed the DEMs and DEGs to gain insights into the mechanism of interaction between the host and the virus (Appendix A).

The overall differences in genes and metabolites between GFabV-infected and GFabV-free berries are presented in Figure 11 (Appendix A). We observed decreases in the expression of several genes in the photosynthesis pathway in GFabV-infected berries, such as photosystem II reaction center protein H (VIT_00s0505g00040), and photosystem II protein D1 (VIT_13s0019g02630). Some defense-related genes were down-regulated by the infection, including disease resistance protein RPS5 (VIT_09s0096g00200), and heat shock protein (VIT_13s0019g02840), while peroxidase 15 (VIT_06s0004g01180) and membrane protein PM19L (VIT_05s0049g02240) were up-regulated.

Some genes involved in the cell wall were activated by GFabV infection, such as pectinesterase 2.2 (VIT_10s0116g00590) and patellin-4 (VIT_10s0003g05480). A large number of amino acids (such as L-cysteine, L-leucine, L-norleucine, and L-isoleucine) were up-regulated as indicated by the metabolome data. In addition, a gene encoding flavonoid 3′,5′-hydroxylase 2 (VIT_06s0009g03050) associated with the flavonoid pathway was positively regulated, suggesting that the flavonoid pathway was also induced by the viral infection. Moreover, the MYB family transcription factor PHL7 (VIT_08s0105g00370), which plays an important role in the flavonoid pathway, was accumulated in infected berries. Some other transcription factors such as bHLH128 (VIT_05s0124g00240) were promoted as well (Appendix A).

### 2.7. Validation of Differential Gene Expression Using RT–qPCR

Changes in the expression of eight DEGs involved in photosynthesis, carbohydrates, hormones, defense, and secondary metabolism were determined in leaves and berries through qRT–PCR. Two reference genes (GAPDH and α-tubulin) were also analyzed to evaluate the gene expression stability in GFabV-infected and free samples. As shown in Figure 12, the expression patterns of these DEGs were consistent with the FPKM values obtained by RNA–seq, representing a high correlation between the RNA–seq and RT–qPCR data and reliability of the sequencing data.

RT–qPCR was also carried out to detect the changes in two genes (photosystem II protein D1 and pathogen-related protein) at four stages (EL15, 19, 27, and 31) (Figure 13).

## 3. Discussion

Our previous studies have demonstrated that GFabV is associated with chlorotic mottling and deformation symptoms in some grapevine samples, with a relatively high incidence in China [6,9]. Therefore, there is an urgent need to further investigate the potential dangers posed by GFabV to grapevines. To gain more insights into GFabV and grapevine interaction, we conducted transcriptomic, metabolomic, physiological, and agronomic studies of both GFabV-infected and GFabV-free ‘Summer Black’ grapevines. This study presents the first description about the influence of GFabV on grapevines.

### 3.1. GFabV Concentration in Infected Grapevines

In this study, we monitored changes in viral concentration in the leaves of ‘Summer Black’ plants at different growth stages throughout the year. Similar to the concentration of grapevine fanleaf virus in Ref DU 2/19 samples [18], the highest GFabV concentration was observed at EL15, while the lowest concentration was found at EL31. Since GFabV and GFLV have similar genomic composition in the same *Secoviridae* family and cause similar symptoms, it can be inferred that the two viruses may have the same or similar biological characteristics.

### 3.2. Physiological and Agronomic Performance of GFabV-Infected Grapevines

GFabV-infected and GFabV-free plants showed significant differences in some physiological indicators and agronomic traits, such as cholorophyll content, P_n_, stomatal conductance, and total cyanine glycosides. GFabV-infected grapevines showed a decrease in P_n_ (17%), which was less dramatic than that observed in ‘Malvasia’ infected with GFLV and GLRaV (45%) [19], and in ‘Nebbiolo’ infected with GVA and GLRaV-3 (60%) [20]. Although P_n_ can directly reflect the photosynthetic capacity [21], it cannot be inferred that the photosynthetic capacity under GFabV infection is lower than that under infection by other grapevine viruses, as photosynthetic capacity is also affected by other factors, such as g_s_, C_i_, transpiration rate, and chlorophyll fluorescence [22]. The attenuation of stomatal conductance would decrease transpiration to limit the transport and absorption of water and nutrients [23], which is consistent with those previous reports in GRSPaV- and GVB-infection grapevines [15,16]. Moreover, GFabV-free grapevines showed progressive accumulation of C_i_, which may be associated with attenuation of the Calvin cycle to affect CO_2_ diffusion through the mesophyll [24]. 

### 3.3. Differences in Aroma Compounds between GFabV-Infected and GFabV-Free Grapevines

In term of grapevine aroma, alcohols and aldehydes were the main aroma compounds in response to GFabV, while alcohols, alkenes, aldehydes and phenols were the main aroma compounds in ‘Summer Black’ grapes [25], suggesting that GFabV infection may not affect the types of aroma compounds of ‘Summer Black’ grapes. However, a comparison of the aroma characteristics with and without GFabV infection revealed a significant difference in phenylethyl alcohol [26], suggesting that GFabV infection affects the aroma characteristics of grape. We speculated that GFabV infection causes changes in the relative contents but not in the types of aroma compounds.

The DEGs at EL15 and EL31 stages were distributed in multiple signaling pathways and biological processes associated with photosynthesis, starch and sucrose metabolism, glycolysis, plant defense, stress response, and hormones. To obtain the key information about these pathways, we would focus on several key aspects related to the pathogenesis of GFabV.

### 3.4. Photosynthesis and Carbohydrate Metabolism in GFabV-Infected Grapevines

The invasion of GFabV can cause chlorotic and necrotic areas on plant leaves, and decrease photosynthetic assimilate production [27]. Moreover, the infection of grapevines would induce some genes involved in CO_2_ fixation [28], with an attempt to increase P_n_. However, these genes involved in CO_2_ fixation are not sufficient to induce an increase in photosynthesis. Similar results were reported in potato by Baebler et al., who observed that the infection of potato virus Y (PVY) led to a transient increase in the expression of photosynthesis-related genes [27].

Plant virus infection can affect plant carbohydrate partitioning and signal transduction, leading to increases in soluble sugars and starch in leaves [29,30], which is likely attributed to the source-to-sink conversion of the infected area. Although the exact mechanism of host-virus interaction remains unclear, it has been suggested that interaction may be associated with the replication of movement proteins (MPs), which may disturb plasmodesmata function or callose deposition, thereby influencing their functions [31,32,33]. Fructose expression was detected at different stages by RT–qPCR and was found to increase in GFabV-infected grapevines. This result supports the hypothesis that GFabV infection inhibits the ‘transport’ process through callose deposition as a defense mechanism, which is consistent with the observations in GVB- and GRSPaV-infected grapevines [15,16].

### 3.5. Changes in Hormone Genes under GFabV Infection in Grapevines

Generally, salicylic acid, jasmonic acid, and ethylene are regarded to be associated with plant defense [34]. These signaling pathways are connected with each other to coordinate modification at the transcriptomic level in response to virus infection with minimal damage [35]. In this study, it seemed that salicylic acid was not implicated in response to GFabV infection. However, jasmonic acid and ethylene signaling were down-regulated in GFabV-infected grapevines. Specifically, jasmonic acid was down-regulated in GFabV-infected leaves at EL31. We speculate that ethylene and jasmonic acid synergistically regulate pathogen response in a salicylic acid-independent pathway, which is similar to the conclusions reported previously [36], but the mechanism for the response remains unclear [37].

### 3.6. Transcription of Defense-Related Genes in Grapevines under GFabV Infection

It was observed that the most significant change under GFabV infection was the down-regulation of defense genes. The genes that perceive specific pathogen effectors to activate effector-triggered immunity, which are called resistance genes (R) such as leucine-rich repeats (LRR), were down-regulated in GFabV-infected plants. Previous studies have suggested that LRR proteins are necessary for plant defense [38], indicating the importance of R genes in this process. In addition, Fortes et al. have reported that grapevine attempts to mediate some genes involved in phenylpropanoid and isoflavonoid biosynthesis in response to fungal infection [39]. Interestingly, our study identified some similar genes as well, such as the gene encoding phenylalanine ammonia-lyase (PAL, VIT_11s0016g01520).

Flavonoids have different effects on biotic and abiotic stress responses [40], and play important roles in plant defense. In this study, we identified some genes involved in flavonoid biosynthesis in leaves, such as butin (EL15), prunetin (EL15), naringin (EL31), and myricetin (EL31). These metabolites may be used as markers of GFabV infection.

### 3.7. Response of GFabV-Infected Grapevine to Several Stresses

Fatty acids and lipids, which are the main constituents of the cell membrane, play significant roles in defense of effector-triggered and systemic immunity [41,42]. Qiao F et al. demonstrated that some genes involved in linoleate 9S-lipoxygenase are related to host-pathogen interaction [43]. In this study, we identified a gene encoding linoleate 9S-lipoxygenase 6 (VIT_07s0141g00340), suggesting an increase in the synthesis of lipids in response to GFabV infection. However, linolenic acid metabolism was negatively regulated at the EL31 stage. We speculate that a novel signal transduction pathway might be activated in response to GFabV, which may be associated with jasmonic acid.

Moreover, glutathione S-transferase and chitinase were identified to be associated with the response to oxidative stress under GFabV infection in this study. As previously reported, glutathione and chitinase play definite roles in biotic and abiotic stress responses [44,45]. However, it remains unclear how chitinase improves plant defense against viruses, though some studies have revealed how it responds to fungi [46,47,48]. Moreover, it is worth noting that some defense genes overlapped in response to GFabV and abiotic stresses. Similar results have been reported in grapevines [15,49]. However, this association remains to be further clarified in future research, particularly in infected leaves and berries. 

### 3.8. Changes in Berries under GFabV Infection

With the development of fruit, a gradual increase in carbohydrate will occur in berries to induce maturation metabolic processes, cell proliferation and expansion, and seed development [50]. In berries, due to the phloem flux limitation, soluble starch content will increase in infected tissues, leading to a decrease in berry weight. Our results were consistent with those observed in GLRaV-3- and GVB-infected grapevines [16,51]. The most significant changes were observed in carotenoids and anthocyanins. As reported, carotenoids can protect cellular structures in response to various stresses in potato [52,53]. In this study, the genes implicated in carotenoid biosynthesis were up-regulated under GFabV infection, suggesting the activation of corresponding pathways. In addition, anthocyanins play a crucial role in host defense against biotic and abiotic stresses [54,55,56]. In this study, the expression of anthocyanidin 3-O-glucosyltransferase (3GT, VIT_11s0052g01630) was down-regulated, while that of Tri-hydrooxylated anthocyanins and flavonoid 3′,5′-hydroxylase (F 3′5′H) was up-regulated in infected berries, which was also reported by Bogs et al. [57]. The reason for these changes may be that an increase in soluble carbohydrates will facilitate anthocyanin biosynthesis and thereby influence their profiles in specific tissues, which is consistent with some reports in grapevines [58,59].

In conclusion, at the EL15 stage, ‘Summer Black’ grapevine leaves had the highest virus concentration and severe chlorotic mottling and deformation symptoms. Under the infection of GFabV, a large number of genes involved in photosynthesis and sucrose synthesis were down-regulated in grapevines, and it was the same case for some genes involved in lipid and secondary metabolism, which is conducive to the invasion of viruses. To resist the infection of the virus, the host would regulate a large number of R genes (such as LRR) and the endogenous genes involved in jasmonic acid and ethylene signaling pathways to protect grapevine growth. Moreover, the TCA cycle was accumulated. It seems that an arm race occurs between the virus and the host. Then, at the EL31 stage, the GFabV concentration was the lowest throughout the year. Although some genes involved in photosynthesis were down-regulated, the plants only showed mild symptoms. In addition, only a few R genes and genes involved in jasmonic acid and ethylene signaling pathways were down-regulated under the viral invasion, and compared with those at the EL15 stage, these genes were very rare, which may be due to the better physiological health of the plants and the transport of virus from leaves to berries at the EL31 stage. Furthermore, when GFabV invaded berries, it also caused the down-regulation of genes involved in the photosystem and changes in R genes, as well as induced changes in carotenoids. The specific molecular mechanism still requires further research.

## 4. Materials and Methods

### 4.1. Plant Materials

Twelve ‘Summer Black’ vines propagated from a single mother plant were planted in pots in May 2016, in a net house in the National Center for Eliminating Viruses from Deciduous Fruit Trees, Research Institute of Pomology, Chinese Academy of Agriculture Sciences (Xingcheng city, Liaoning province, China) (Appendix A). After these plants survived, six grapevines were inoculated with GFabV by grafting a bud from a GFabV-infected ‘Beta’ grapevine. Before grafting, the ‘Summer Black’ mother vines and GFabV-infected Beta grapevines were repeatedly tested for grapevine viruses reported in China, including grapevine leafroll-associated virus-1, -2, -3, -4, -7, and -13, grapevine rupestris stem pitting-associated virus, grapevine fleck virus, grapevine fanleaf virus, grapevine virus A, grapevine virus B, grapevine virus E, GPGV, GFabV, grapevine red blotch virus, grapevine rupestris vein feathering virus, grapevine Syrah virus-1, and grapevine red globe virus. The ‘Summer Black’ mother plants were tested to be negative for all these viruses, and only the ‘Beta’ grapevine was tested to be positive for GFabV. Chlorotic mottling symptoms could be observed in GFabV-inoculated ‘Summer Black’ grapevine leaves two months after the grafting, while the leaves of uninoculated ‘Summer Black’ grapevine showed no symptoms. In 2017, it was further confirmed by RT–PCR that GFabV infected the inoculated grapevines but not the uninoculated grapevines. 

Grapevine leaves only infected with GFabV were observed by visual inspection at the EL15 and the EL31 stages (7 May and 12 July 2019). At the EL15 stage, the leaves showed chlorotic mottling and deformation symptoms (Appendix A). With the growth of plants, the symptoms became less obvious, and at the EL31 stage, the leaves showed no significant symptoms (Appendix A). It should be noted that the EL system modified by Coombe [60] represents different phenological phases, including the eight-leaf period (EL15), flowering initiation (EL19), berry setting (EL27), pea sized berries (EL31), véraison (EL35), and harvest period (EL38).

### 4.2. Agronomical and Physiological Parameters

Some agronomic parameters were determined at the harvest stage (EL38) in 2018 and 2019. The bunch weight (cm), bunch width (cm), and bunch length (cm) of the dormant bunches were recorded. Based on the methods of the International Organization of Vine and Wine (http://www.oiv.int/oiv/info/frmethodesinternationalesvin, accessed on 10 March 2022), berry weight (g), length (cm), width (cm), soluble solids (°Brix), titratable acidity (g/L), and total cyanine glycosides (mg/100 g) at LE38 were recorded. The physiological and agronomic data collected over these two years were statistically analyzed with the analysis of variance (ANOVA) *F*-test.

During the 2018 and 2019 plant growth periods, the main physiological parameters, including photosynthetic rate (P_n_), transpiration rate (E), stomatal conductance (g_s_), and substomatal CO_2_ concentration (c_i_) were detected using the LI-6400 Portable Photo Synthesis Measurement System (LI-COR, LincolnNE, USA) in GFabV-infected and uninfected ‘Summer Black’ vines. The chlorophyll content was monitored by a non-destructive portable chlorophyll meter SPAD-502 (Konica Minolta, Tokyo, Japan). These data were recorded from 12 am to 4 pm on three leaves per plant at the EL15 and EL31 stages. 

### 4.3. GC–MS Analysis

The volatile composition was quantified in duplicate by headspace-solid phase microextraction (HS–SPME) coupled to gas chromatography (GC) with mass spectrometry (MS) according to Wang et al. [61] with minor modifications. A GC–MS–QP2010 instrument was used to perform GC–MS analysis. An ADB-5MS column (30 m × 0.32 mm, 0.5 µm) (J&W Scientific, Folsom, CA, USA) was used for all analyses. The oven parameters were as follows: the initial temperature was 50 °C, held for 2.0 min, followed by an increase to 150 °C at a rate of 1.0 °C min^−1^, immediately increased to 180 °C at a rate of 2 °C min^−1^, and finally increased to 230 °C at a rate of 10 °C min^−1^, and then held for 10 min. High-purity helium (purity 99.999%) was used as the carrier gas with a flow rate of 1 mL min^−1^. Samples were injected in the splitless mode. The Agilent 5975C MS was operated in the electron impact (EI) mode using ionization energy at 70 eV with an ionization source temperature of 200 °C. The acquisition mode was full scan (from 50 to 500 m/z) and the solvent delay time was 2.8 min.

These volatile compounds were qualitatively analyzed by the m/z values, the RT (retention time), the fragmentation patterns with the standards, the National Institute of Standards and Technology (NIST) 17 library Mass Spectra Database, and retrieving the mass spectrometry data of each component with reference to relevant literature. The relative percentage contents of each compound were calculated through the peak area normalization method.

### 4.4. RNA–Seq Experiment and Data Analysis

RNA–seq analysis was performed on RNA extracted from leaves at EL15 and EL31 and berries at EL38 in 2019. Three GFabV-free and three GFabV-infected mixed leaves (basal and apical) at EL15 and EL31 were randomly collected, and berries were collected from three different bunches. The RNA quality and quantity were measured by both the Nanodrop (Life Technologies, Carlsbad, CA, USA) and a Bioanalyzer 2100 (Agilent Technologies, Santa Clara, CA, USA). mRNA was obtained from total RNA by poly-Toligo-attached magnetic beads. Then, the mRNA molecules were fragmented, and applied to the first- and second-strand complementary (cDNA) synthesis. The cDNA was subsequently subjected to end repair, and poly (A) and unique adapter ligation. Before sequencing, the cDNA fragments were amplified and purified. The purified and amplified products were sequenced on the Illumina novaseq 6000 platform. The raw sequencing data were defined as raw reads. After removal of low-quality reads, mismatches, and adaptor sequences, clean reads were generated from the raw reads. The reference genome (International grapes genomes program, IGGP, www.vitaceae.org, accessed on 12 December 2020, 12x. 45 version) was used. Moreover, the Q20, Q30, GC content, and sequence duplication level of the clean data were calculated. All downstream analyses were based on clean data with high quality.

Differential gene expression analysis was carried out by the DESeq2_EBSeq package and *p* values were adjusted with the Benjamini-Hochberg method for controlling the false discovery rate (FDR). The differentially expressed genes (DEGs) were screened by the following thresholds: log_2_|fold change| > 1.5 and *p* value < 0.05. Then, DEGs were annotated by matching against the non-redundant protein sequence (NR), Swiss-Prot, Kyoto Encyclopedia of Genes and Genomes (KEGG), Clusters of Orthologous Groups of proteins (COG), Eukaryotic Orthologous Groups (KOG), Gene Ontology (GO), and Pfam databases.

### 4.5. Real-Time RT–PCR

Validation of the RNA–seq data was performed in biological triplicates on the same RNA samples subjected to RNA–seq analysis. RNA was extracted as described by Fan et al. [62], and the RNA quality and quantity were checked. Following the instructions of the PrimeScriptTMRT reagent Kit with gDNA Eraser (TaKaRa, Beijing, China), 1 μg of template RNA for cDNA analysis was added to 20 mL mixture and 10-fold diluted for qRT–PCR analysis. Primers (Appendix A) were designed by Primer Express^®^ 6.0 software (Applied Biosystems, Foster City, CA, USA) and qRT–PCR was performed using the CFX ConnectTM Real-Time System (BIO-RAD, Hercules, CA, USA), as previously described by Zhang et al. [63]. The relative expression levels of the target genes were calculated using the 2^−ΔΔCT^ method with GAPDH and α-tubulin as the reference gene [64].

To better understand the changes of some genes in leaves at different stages (EL15, 17, 27, and 31), we used the above method [63] to determine these genes by RT–qPCR.

### 4.6. UPLC–MS/MS Analysis for Widely-Targeted Metabolome Study

The metabolites’ extraction and widely targeted metabolomics were both performed by the Biomarker Technologies Corporation (Beijing, China). The samples were analyzed using an UPLC-ESI-MS/MS system (UPLC, SHIMADZU Nexera X2, www.shimadzu.com.cn, accessed on 10 March 2022; MS, Applied Biosystems 4500 Q TRAP, www.appliedbiosystems.com.cn, accessed on 10 March 2022). The analytical conditions were as follows: UPLC column, Agilent SB-C18 (1.8 µm, 2.1 mm × 100 mm). The mobile phase consisted of solvent A with pure water and 0.1% formic acid, and solvent B with acetonitrile and 0.1% formic acid. The sample measurements were carried out with a gradient program under the starting conditions of 95% A and 5% B. Within 9 min, a linear gradient to 5% A and 95% B was programmed, and a composition of 5% A and 95% B was maintained. Then, composition of 95% A and 5.0% B was adjusted within 1.10 min and kept for 2.9 min. The column oven temperature was set to 40 °C with an injection volume of 4 µL. The effluent was alternatively connected to an ESI-triple quadrupole-linear ion trap (QTRAP)-MS.

LIT and triple quadrupole (QQQ) scans were acquired on a triple quadrupole-linear ion trap mass spectrometer (Q TRAP), AB4500 Q TRAP UPLC/MS/MS System, equipped with an ESI Turbo Ion-Spray interface, operated in the positive and negative ion mode and controlled by Analyst 1.6.3 software (AB Sciex). The ESI source operation parameters were as follows: ion source, turbo spray; source temperature, 550 °C; ion spray voltage (IS), 5500 V (positive ion mode)/−4500 V (negative ion mode); ion source gas I (GSI), gas II (GSII); the curtain gas (CUR) was set at 50, 60, and 25.0 psi, respectively; and the collision gas (CAD) was high. Instrument tuning and mass calibration were performed with 10 and 100 μmol/L polypropylene glycol solutions in QQQ and LIT modes, respectively. QQQ scans were acquired as MRM experiments with collision gas (nitrogen) set to medium. DP and CE for each MRM transition were performed with further optimization. A specific set of MRM transitions were monitored for each period according to the metabolites eluted during this period.

Based on the local metabolic database, the substance was characterized according to the secondary spectral information, and the isotopic signals, repeated signals containing K^+^, Na^+^, NH4^+^, and fragment ions that were themselves larger molecular weight substances were removed. Metabolite quantification was performed using the MRM of triple quadrupole mass spectrometry. In the MRM mode, the quadruple first screens the precursor ions (precursor ions) of the target substance, and excludes the ions corresponding to other molecular weight substances to preliminarily eliminate interference. The precursor ions are induced by the collision chamber and then fractured to form many fragment ions, and the fragment ions are then filtered through the triple quadruple rod to select a characteristic fragment ion required to eliminate the interference of non-target ions, so that the quantification is more accurate and the repeatability is better. After obtaining the metabolite profiling data of different samples, the peak area is integrated for all mass spectral peaks of all substances, and the mass spectrum peaks of the same metabolite in different samples are integrated and corrected [65].

The metabolomics data were identified based on the BioMarker Technologies Corporation self-built database (www.bi-omarker.com.cn, accessed on 10 March 2022) and several publicly available metabolite databases, namely HMDB (www.hmdb.ca, accessed on 10 March 2022), MassBank (www.massbank.jp, accessed on 10 March 2022), MoToDB (www.ab.wur.nl/moto, accessed on 10 March 2022), and METLIN (metlin.scripps.edu/index.php). The filtered data were submitted to Simca-P software (version 14.0, Umetrics AB, Umea, Sweden) for unsupervised principal component analysis (PCA) or orthogonal partial least squares-discriminant analysis (OPLS-DA). Hierarchical clustering analysis of the metabolites between the samples was performed using R software (www.r-project.org, accessed on 10 March 2022). For identifying the differentially expressed metabolites (DEMs), the screening criteria were *p* value ≤ 0.05, fold change ≥ 1, and variable importance in project (VIP) ≥ 1.

## Figures and Tables

**Figure 1 ijms-24-03247-f001:**
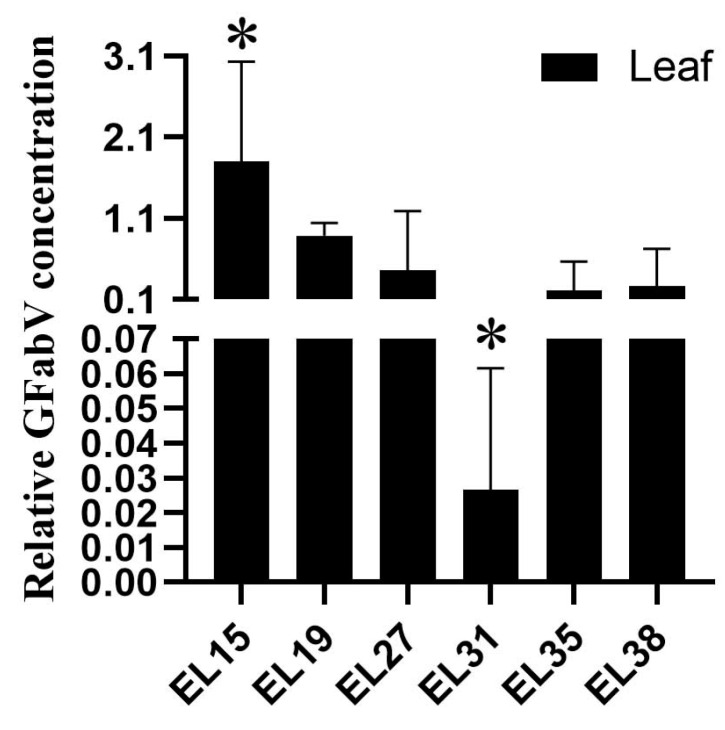
Quantification of grapevine fabavirus (GFabV) RNA in leaves of ‘Summer Black’ as determined by RT–qPCR. Samples were collected from six developmental stages in 2019 (EL15, 19, 27, 31, 35, and 38). Data are expressed as mean ± SE of biological and technical replicates. Bars represent the standard error of the mean. Asterisks indicate significant differences between GFabV-infected and GFabV-free leaves. (*p* < 0.05).

**Figure 2 ijms-24-03247-f002:**
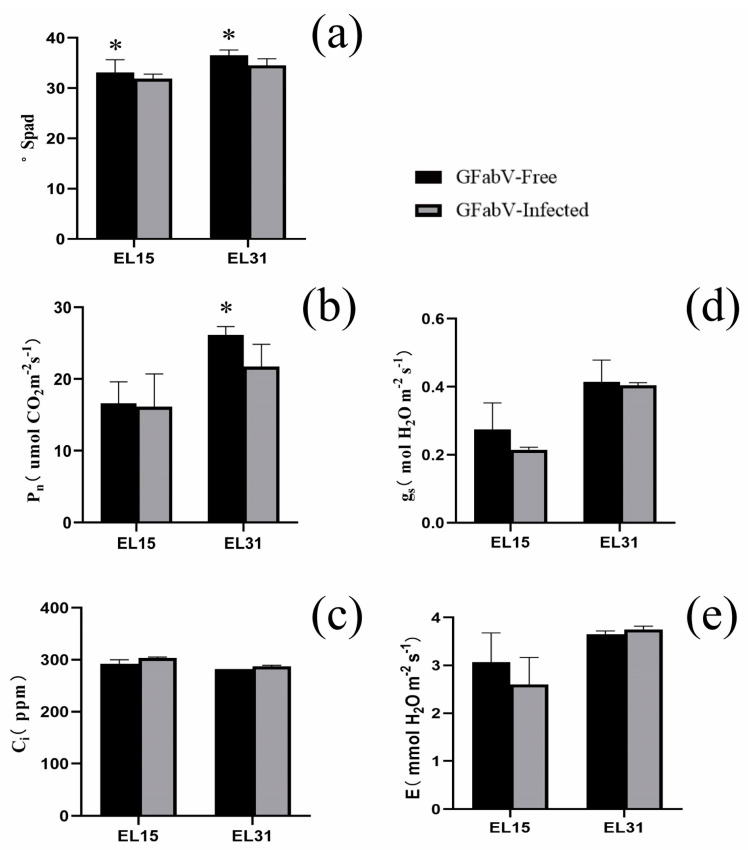
Comparison of (**a**) chlorophyll content (Spad), (**b**) net photosynthesis (P_n_), (**c**) intercellular CO_2_ concentration (c_i_), (**d**) stomatal conductance (g_s_), and (**e**) transpiration (E) in GFabV-free (black columns) and GFabV-infected (grey columns) in grapevine leaves in 2019. Measurements (n = 3) were conducted in two phenological periods: EL15 and EL31. Bars represent the standard error (SE). Statistically significant results are indicated with asterisks, *p* < 0.05.

**Figure 3 ijms-24-03247-f003:**
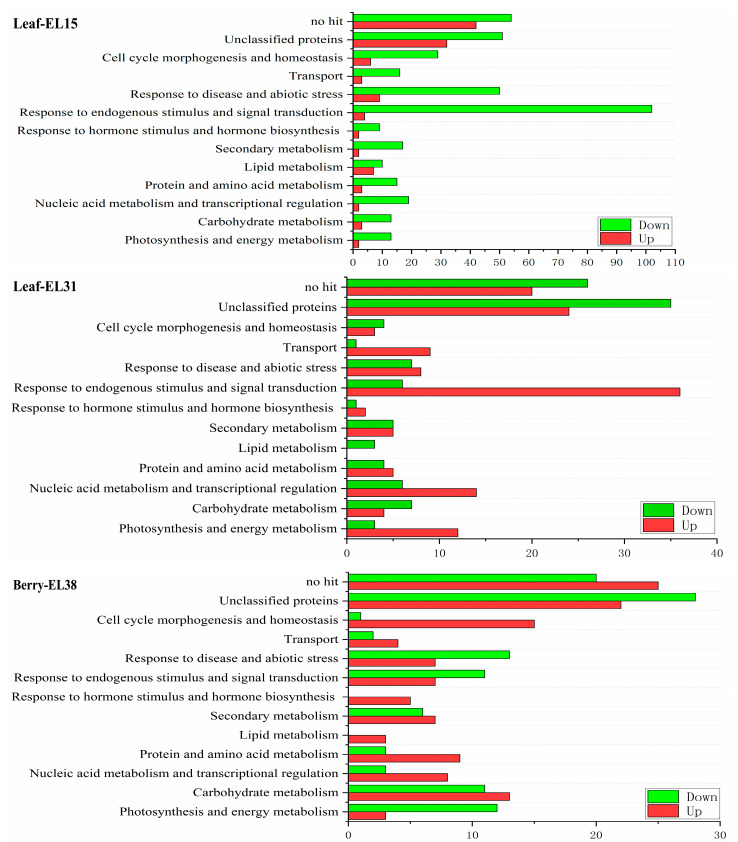
Functional assignment of transcripts significantly promoted or inhibited in GFabV-infected leaves (EL15, EL31) and berry (EL38) samples in 2019. Bars indicate the up-regulated (red) or down-regulated (green) genes in related categories.

**Figure 4 ijms-24-03247-f004:**
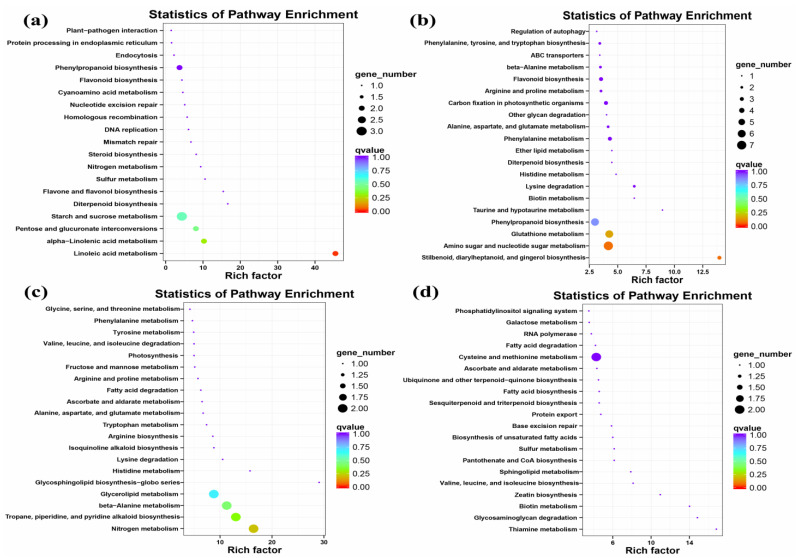
Pathways analysis of differentially expressed genes in leaf. (**a**) KEGG enrichment analysis of up-regulated genes at EL15; (**b**) down-regulated genes at EL15; (**c**) up-regulated genes at EL31; (**d**) down-regulated genes at EL31. The size of Bubbles represents the amount of genes, and the color represents the qvalue.

**Figure 5 ijms-24-03247-f005:**
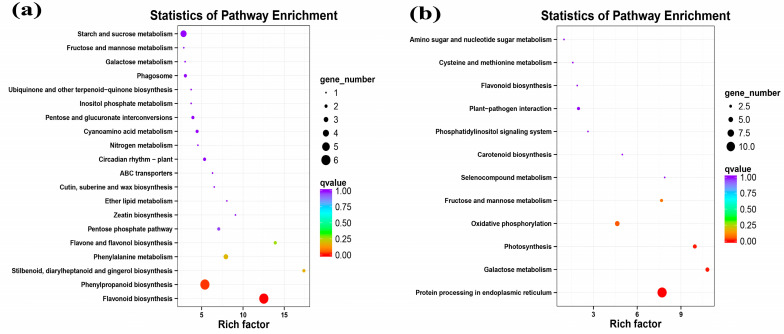
Gene expression profiles of ‘Summer Black’ grapevine berries. (**a**) Top 20 pathways on the basis of KEGG pathway enrichment analysis of up-regulated genes at EL38. (**b**) Pathway enrichment analysis of down-regulated genes at EL38. The size of bubbles represents the number of genes, and the color represents qvalue.

**Figure 6 ijms-24-03247-f006:**
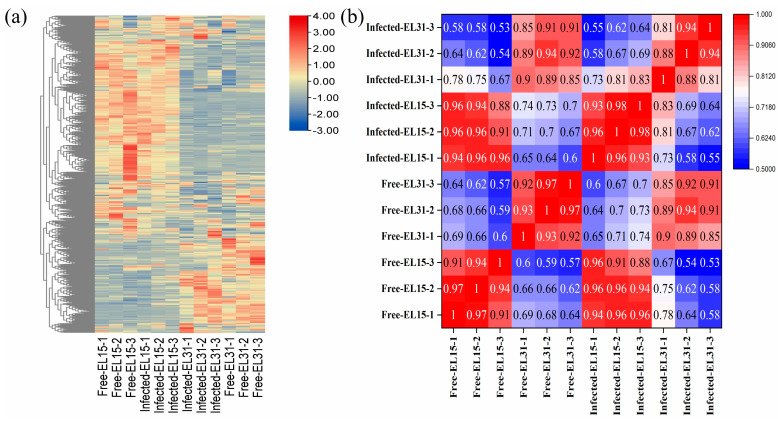
Multivariate statistical analysis of metabolites in leaves. (**a**) Heatmap of the 595 metabolites. Red represents up-regulation, and blue represents down-regulation; (**b**) Pearson correlation coefficients of the twelve leaf samples.

**Figure 7 ijms-24-03247-f007:**
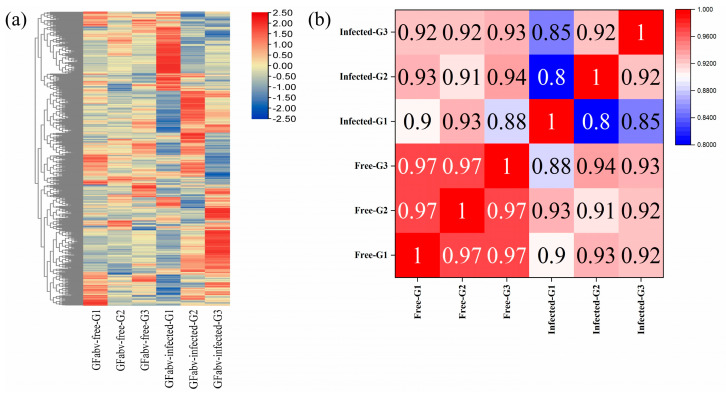
PCA, heatmap, and correlation analysis of metabolites in berries. (**a**) Clustering heat map of all metabolites. Red represents up-regulation, and blue represents down-regulation; (**b**) Pearson correlation coefficients of the six berry samples.

**Figure 8 ijms-24-03247-f008:**
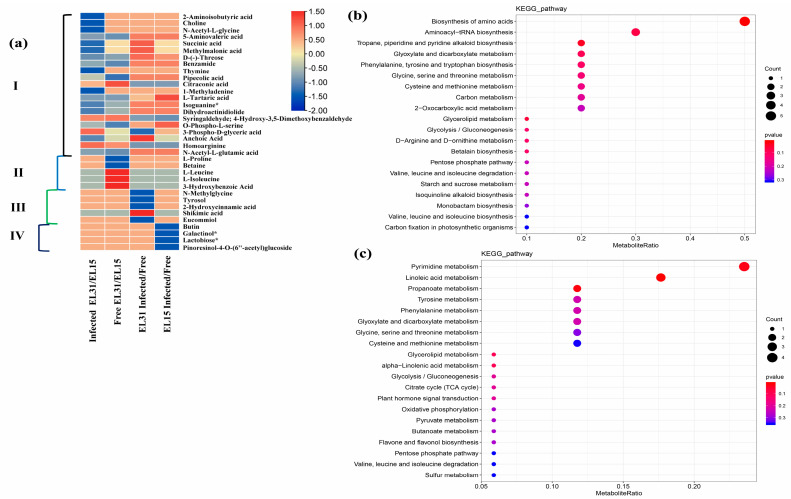
Analysis of differentially expressed metabolites (DEMs) in leaves. (**a**) Metabolites with significant increases or decreases in at least one of the comparisons. The four clusters(I–IV) represent the respective response patterns (Appendix A). A small fraction of metabolites was selected from each cluster. (**b**) Top 20 enrichment pathways on the basis of KEGG analysis of the total DEMs between GFabV-infected and GFabV-free leaves at EL15; (**c**) KEGG annotation of top 20 DEMs at EL31. The size of bubble represents the number of metabolites in the pathway, the color represents the *p* value.

**Figure 9 ijms-24-03247-f009:**
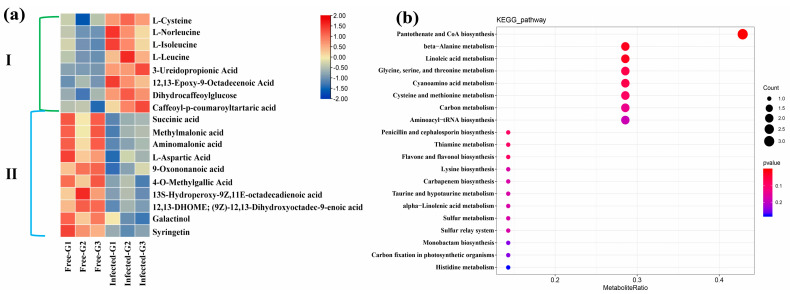
Profiling of differentially expressed metabolites (DEMs) in berries. (**a**) Heatmap of DEMs. I represents the up-regulated metabolites, and II indicates the down-regulated metabolites. (**b**) Top 20 enrichment pathways based on KEGG analysis of the total DEMs between GFabV-infected and -free berries. The size of bubble represents the number of metabolites in the pathway, and the color represents the *p* value.

**Figure 10 ijms-24-03247-f010:**
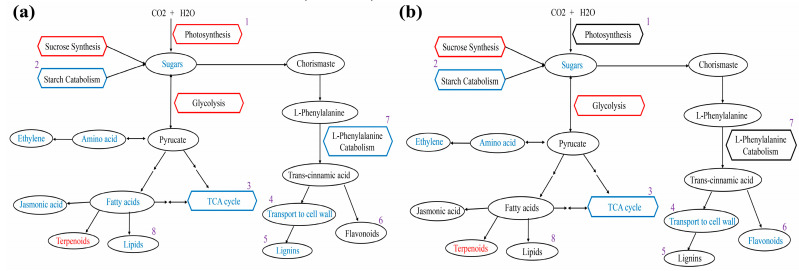
Analysis of transcriptomic-metabolomic data between GFabV-infected and GFabV-free in leaves. (**a**) Overview of the major changes in response to GFabV in ‘Summer Black’ at EL15. Ovals and hexagons represent metabolites and pathways, respectively. Red represents down-regulated genes and metabolites, whereas blue represents up-regulated genes and metabolites. The numbers (1–8) represent the pathways analyzed on the basis of Appendix A. (**b**) Overview of the major changes in response to GFabV in ‘Summer Black’ at EL31. The numbers (1–8) represent the pathways analyzed on the basis of Appendix A.

**Figure 11 ijms-24-03247-f011:**
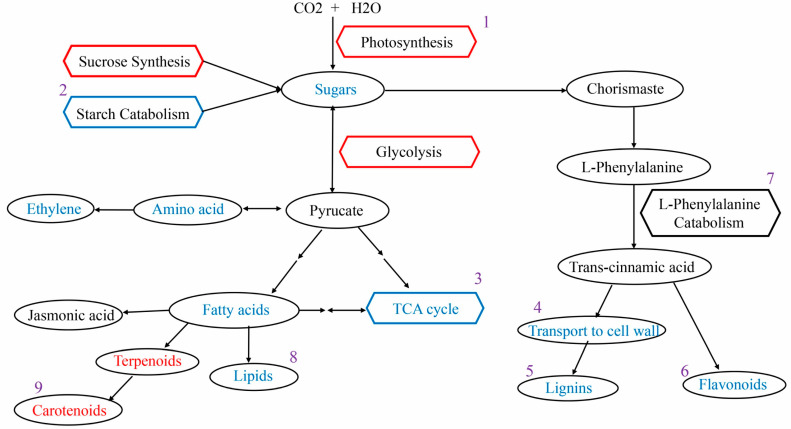
Profiling of transcriptomic-metabolomic data between GFabV-infected and -free berries. Overview of the major changes in response to GFabV in ‘Summer Black’. Ovals and hexagons represent metabolites and pathways, respectively. Red represents down-regulated genes and metabolites, while blue represents up-regulated genes and metabolites. The number (1–9) represents the pathways analyzed on the basis of Appendix A.

**Figure 12 ijms-24-03247-f012:**
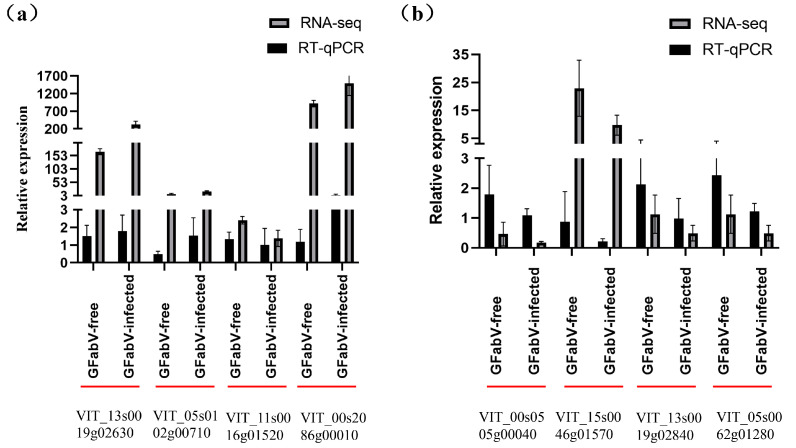
Validation of the expression of genes in ‘Summer Black’ (**a**) berries and (**b**) leaves by RT–qPCR. Error bars represent the standard deviation of three independent biological repeats. Statistically significant results are indicated as *p* < 0.05.

**Figure 13 ijms-24-03247-f013:**
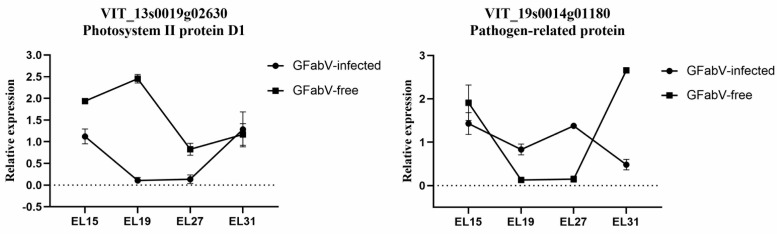
Relative expression levels of important genes in GFabV-free and GFabV-infected ‘Summer Black’ leaves analyzed by RT–qPCR. Grapevine leaves were collected at four stages (EL15, 19, 27, and 31) in 2019. Data are shown as mean ± SE for biological and technical replicates.

**Table 1 ijms-24-03247-t001:** Agronomic traits and Fruit quality indicators of GFabV-infected and GFabV-free of ‘Sum mer Black’ grapevines in two consecutive years.

Data	2018			2019		
	GFabV-Free	GFabV-Infected	*p*-Value	GFabV-Free	GFabV-Infected	*p*-Value
Berry weight (g)	10.38 ± 0.08	9.14 ± 0.33	**	7.67 ± 0.98	5.33 ± 1.03	NS
Berry length (cm)	27.6 ± 0.13	26.64 ± 1.41	*	16.24 ± 0.34	14.60 ± 0.14	**
Berry width (cm)	27.53 ± 0.23	26.92 ± 1.16	**	16.63 ± 0.86	15.23 ± 0.95	NS
Soluble solids (°Brix)	18.43 ± 0.06	18.4 ± 0.26	NS	18.60 ± 0.41	17.63 ± 1.05	NS
Titratable acidity (g/L)	0.31 ± 0.001	0.34 ± 0.01	*	0.43 ± 0.08	0.48 ± 0.05	NS
Total cyanine glycosides (mg/100 g)	114.9 ± 6.38	191.2 ± 2.38	**	112.93 ± 2.82	191.79 ± 1.87	**
Bunch weight (g)	253 ± 8.89	290 ± 13.5	NS	506.7 ± 56.62	440 ± 56.57	NS
Bunch length (cm)	11.33 ± 0.58	9.33 ± 2.52	NS	14 ± 1.63	12.33 ± 0.47	NS
Bunch width (cm)	0.47 ± 0.06	0.6 ± 0.10	NS	1 ± 0.08	0.73 ± 0.12	NS

All data are expressed as average values ± SE. * *p* ≤ 0.05; ** *p* ≤ 0.01; NS, not significant.

**Table 2 ijms-24-03247-t002:** Screening of genes significantly regulated in GFabV-infected and GFabV-free Leaves (at EL15, EL31) and Berries (EL38) in 2019 (considering a *p* value < 0.05 and log_2_(fold change) ≥ 1.5 or ≤1.5).

Unique ID	Functional Annotation	Log_2_(Fold Change)		
		Leaf		Berry
		EL15	EL31	EL38
Photosynthesis and energy metabolism				
VIT_18s0001g11470	Cytochrome P450 82A3	−0.86		
VIT_00s0505g00040	Photosystem II reaction center protein H			−1.45
VIT_07s0141g00890	Cytochrome P450 94A1		0.85	
VIT_13s0019g02630	Photosystem II protein D1		0.82	−1.92
VIT_18s0122g00460	Calvin cycle protein CP12–3, chloroplastic		0.64	
VIT_08s0007g07530	NADPH-dependentaldehyde reductase-like protein, chloroplastic	−1.09		
Carbohydrate metabolism				
VIT_06s0061g00360	UDP-glycosyltransferase 86A1	−0.67		
VIT_05s0062g01280	UDP-glycosyltransferase 92A1			−1.15
VIT_11s0052g01190	Xyloglucanendotransglucosylase/hydrolase protein 23			1.14
VIT_03s0038g04570	Glucose-1-phosphate adenylyltransferase large subunit 1 (Fragment)		−1.58	
VIT_05s0062g00240	Xyloglucan endotransglucosylase/hydrolase 2	0.62		
VIT_00s0233g00030	Trehalose-phosphate phosphatase F		0.69	
VIT_05s0062g00250	Xyloglucan endotransglucosylase/hydrolase 2			1.81
VIT_05s0102g00710	Fructose-2		0.71	
VIT_14s0108g00890	Beta-1,3-galactosyltransferase GALT1	−0.66		
Nucleic acid metabolism and transcriptional regulation				
VIT_08s0058g00690	Probable WRKY transcription factor 33	−0.61		
VIT_08s0105g00370	MYB family transcription factor PHL7			1.13
VIT_08s0007g05030	Transcription factor MYB36		0.71	
VIT_06s0004g04140	Transcription factor MYB59	−0.64		
VIT_18s0001g10030	WRKY transcription factor 7		0.63	
VIT_05s0124g00240	Transcription factor bHLH128			1.14
VIT_01s0011g03070	AP2/ERF and B3 domain-containing transcription repressor TEM1		0.62	
Lipid metabolism				
VIT_11s0016g00120	Omega-6fattyaciddesaturase, chloroplastic	0.62		
VIT_07s0141g00340	Linoleate 9S-lipoxygenase 6 (Fragment)	0.63		
VIT_00s0583g00030	Sphingolipiddelta (4)-desaturaseDES1-like		−0.72	
VIT_11s0052g01090	4-coumarate—CoA ligase 1			1.03
VIT_02s0109g00250	4-coumarate—CoA ligase-like 6		−0.63	
VIT_14s0171g00110	Triacylglycerol lipase 2	0.81		
Secondary metabolism				
VIT_11s0016g01520	Phenylalanine ammonia-lyase	−0.61		
VIT_06s0009g03050	Flavonoid 3′,5′-hydroxylase 2			0.91
VIT_11s0065g00130	Beta-amyrin 28-oxidase	−0.69		
VIT_06s0004g06130	Phosphomethylpyrimidinesynthase, chloroplastic		−0.73	
VIT_16s0100g01200	Stilbene synthase 6			2.19
VIT_02s0025g04880	Geraniol 8-hydroxylase		0.59	
VIT_06s0009g02910	Flavonoid 3′,5′-hydroxylase 2	0.66		
VIT_18s0001g03470	Flavonolsynthase/flavanone 3-hydroxylase			−1.41
VIT_03s0038g04620	Isoflavone reductase homolog PCBER	−0.78		
Response to hormone stimulus and hormone biosynthesis				
VIT_16s0013g01000	Ethylene-responsive transcription factor 5	−0.66		
VIT_06s0004g06790	Gibberellin 2-beta-dioxygenase 8		−0.59	
VIT_05s0020g04680	Auxin-induced protein 22D	−0.80		
VIT_17s0000g02420	Auxin efflux carrier component 1			1.07
VIT_00s2086g00010	1-aminocyclopropane-1-carboxylate oxidase	−0.67	0.59	
VIT_17s0000g06210	Gibberellin-regulated protein 6			1.39
VIT_10s0116g00410	Gibberellin 2-beta-dioxygenase 8	0.65		
Response to endogenous stimulus and signal transduction				
VIT_10s0003g02050	Cyclic phosphodiesterase	−0.70		
VIT_09s0002g07720	Receptor-like protein 1		0.66	
VIT_19s0014g00520	Leucine-rich repeat receptor-like serine/threonine-protein kinase At3g14840			1.02
VIT_16s0050g01530	Receptor-like protein EIX1	−0.74		
VIT_12s0034g01300	TMV resistance protein N	−0.74		
VIT_14s0108g01000	Calcium-binding protein CML45			−0.90
VIT_00s0125g00090	LRR receptor-like serine/threonine-protein kinase			
	At1g56130	−0.62	0.70	
Response to disease and abiotic stress				
VIT_13s0019g02840	18.2 kDa class I heat shock protein			−1.16
VIT_04s0008g00150	NAC domain-containing protein 22	−0.79		
VIT_06s0004g07750	Cationic peroxidase 1	0.69		
VIT_09s0096g00200	Disease resistance protein RPS5			−1.03
VIT_03s0091g00560	Leucine-rich repeat protein 2	−0.67		
VIT_17s0000g02950	Glutathione S-transferase	−0.89		
VIT_06s0004g03120	MLO-like protein 3		0.62	
VIT_04s0023g02570	Peroxidase 72	−1.01		
VIT_02s0025g00280	Heat shock protein 83	0.70		
VIT_02s0012g01040	NAC domain-containing protein 71		−0.76	
VIT_15s0046g01570	Acidic endochitinase			−1.25
VIT_06s0004g01180	Peroxidase 15			1.01
VIT_13s0067g03220	ProteinENHANCEDDISEASE RESISTANCE 4	−0.66		
Transport				
VIT_05s0020g00890	ABC transporter B family member 11	−0.66		
VIT_14s0066g01000	Phosphoenolpyruvate/phosphate translocator 2, chloroplastic		0.60	
VIT_17s0000g02500	Ras-related protein RABA6a			−0.91
VIT_01s0127g00070	High affinity nitrate transporter 2.5		1.02	
VIT_03s0063g02250	Polyol transporter 5			2.15
VIT_03s0017g02170	Zinc transporter 5	−0.63		
VIT_04s0008g04180	Silicon efflux transporter LSI2	−0.61		
Cell cycle, morphogenesis, and homeostasis				
VIT_00s0414g00010	Cellulose synthase-like protein E6	−0.63		
VIT_11s0016g01530	Ankyrin repeat-containing protein BDA1	−0.68		
VIT_15s0021g02680	Cyclin-U2-1			1.71
VIT_06s0009g02020	Tubulin beta-6 chain		−0.63	
VIT_02s0025g01850	Cellulose synthase-like protein G3		0.60	
VIT_03s0017g01010	Metacaspase-1	0.74		
VIT_10s0003g05480	Patellin-4			2.02

## Data Availability

All data supporting the findings and conclusions of this work are available in the manuscript. Appendix A will be made available from the corresponding author upon request.

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
