# Peer review of "Integrated Transcriptome and Metabolome Dissecting Interaction between Vitis vinifera L. and Grapevine Fabavirus"

_ijms, 2023, doi:10.3390/ijms24043247_

Round 1

Reviewer 1 Report

The manuscript about the transcriptome and metabolome analysis of grapevine plants affected by GFabV has a discrete potential in the  description of collected data. However it suffers, mainly in the Discussion section , of a rather quick elaboration and an overlapping of findings without a fluent and convincing  construction of relationship between causality, evidences and effects. In my view, Discussion seems more a list of findings than a critical analysis of them, and by the way it should be re-written with some synthesis and downsizing. As well, the ampelometric measures performed did find a poor evaluation in the comparison of healthy and infected plants. Finally, I wonder about the total absence of a grapevine background virome in the tested accession: I do not know the history of this bred grapevine cultivar , but it could be really the first time for a grapevine source to be completely free of viruses and viroids  (if it does not come from meristem tip culture and  sanitation procedures)

English spelling and syntax should be deeply and thoroughly revised , as well as the amount of mistyping and incorrect verb conjugation.

Specific points to be checked:

line 42, specify that Shine Muscat is a hybrid crossing between two cultivars

eliminate abbreviated personal names from citations 

103, which differential symptoms ? on which organ?

155-156, rephrase the sentences and explain what gene is not annotated and why could it be important since unknown.

in Fig. 4A it seems to me that pictures ( 'infection' and  'stage' response) are inverted 

in Tab.3 an E-L35 period is reported (erroneously ? ) for berries , while the legend tells EL38

in Fig. 8 legend , informative sentences about points (b) and (c) are inverted (?)

227, Pearson correlation (?)

262, good separation

I do not find any explanation  in the text of  Fig12 in the paragraph Validation of differential gene expression using RT-qPCR but only to fig. 13

428, fig 13  shows only two analysed genes ;  remove from this section the sentence "the results were conducive to the understanding of the host-virus mechanism". 

440, prevalence, better than incidence

489, signaling

514-518 , rephrase entire sentence since it is difficult in understanding

544, early stages

564, tested, better than detected

577, showed, better than behaved

585, potted grapevines are cultivated under screenhouse , not in the real field condition

607, Wang 

634 and 126, it is not clear how the mapping against the reference genome has been done. please support with more details on the procedure

In Results, the % obtained from the mapping is not reported; also it is not clear if the transcriptome ( of a different cv compared to the reference genome) was de novo assembled or not, and how this has been done. This is a crucial point 

637, data is already a plural noun

649, confirm if it was necessary a DNAse digestion of the purified RNA used in qPCR to remove residual genomic DNA (it could be present as low contamination even after RNA magnetic beads capture)

659, which kind of extraction did the samples undergo to ?

743, check the reference nr. 7 since the authors given names are exposed and not family names

Author Response

The manuscript about the transcriptome and metabolome analysis of grapevine plants affected by GFabV has a discrete potential in the description of collected data. However, it suffers, mainly in the Discussion section, of a rather quick elaboration and an overlapping of findings without a fluent and convincing construction of relationship between causality, evidences and effects. In my view, Discussion seems more a list of findings than a critical analysis of them, and by the way it should be re-written with some synthesis and downsizing. As well, the ampelometric measures performed did find a poor evaluation in the comparison of healthy and infected plants. Finally, I wonder about the total absence of a grapevine background virome in the tested accession: I do not know the history of this bred grapevine cultivar, but it could be really the first time for a grapevine source to be completely free of viruses and viroids (if it does not come from meristem tip culture and sanitation procedures)

Response: Thank you!In accordance with your comments, we have made careful revisions., especially the discussion section.  The tested ‘summer black’ vines propagated from a single mother plant,which come from eristem tip culture and sanitation procedures. The samples tested negative for all other viruses, and only tested positive for GFabV in infected samples.

English spelling and syntax should be deeply and thoroughly revised, as well as the amount of mistyping and incorrect verb conjugation.

Specific points to be checked:

line 42, specify that Shine Muscat is a hybrid crossing between two cultivars; eliminate abbreviated personal names from citations 

Response: We added the parent information about Shine Muscat, and eliminate abbreviated personal names from citations. Thank you!

103, which differential symptoms ? on which organ?

Response: Sorry, to avoid ambiguity, we modified the statement.

155-156, rephrase the sentences and explain what gene is not annotated and why could it be important since unknown.

Response: We have rephrased this sentence.

in Fig. 4A it seems to me that pictures ( 'infection' and  'stage' response) are inverted 

Response: Thank you for your question! In Fig.4a, the first figure showed the infection-responsive transcripts between two different stage, and the second figure showed the two stage-responsive transcripts between free and infected plants.

in Tab.3 an E-L35 period is reported (erroneously ? ) for berries , while the legend tells EL38

Response: Sorry, it was a clerical error, should be EL38, we have changed it. Thank you!

in Fig. 8 legend , informative sentences about points (b) and (c) are inverted (?)

Response: Yes, we have corrected the mistake. Thank you!

227, Pearson correlation (?)

Response: Thank you! We have corrected it.

262, good separation

Response: Thank you! We have corrected it.

I do not find any explanation  in the text of  Fig12 in the paragraph Validation of differential gene expression using RT-qPCR but only to fig. 13

Response: Thank you!We added the Fig 12 annotation in the text.

428, fig 13  shows only two analysed genes ;  remove from this section the sentence "the results were conducive to the understanding of the host-virus mechanism". 

Response: Thank you!We have remove these.

440, prevalence, better than incidence

Response: Thank you! We have changed “incidence” to “prevalence”.

489, signaling

Response: Thank you! We have corrected it.

514-518 , rephrase entire sentence since it is difficult in understanding

Response:Thank you! We have rewrited it.

544, early stages

Response: Thank you! We have corrected it.

564, tested, better than detected

Response: Thank you! We have corrected to ‘tested’.

577, showed, better than behaved

Response: Thank you! We have corrected it.

585, potted grapevines are cultivated under screenhouse , not in the real field condition

Response: Thank you! We have corrected this mistake.

607, Wang 

Response: Thank you! We have corrected it.

634 and 126, it is not clear how the mapping against the reference genome has been done. please support with more details on the procedure

In Results, the % obtained from the mapping is not reported; also it is not clear if the transcriptome ( of a different cv compared to the reference genome) was de novo assembled or not, and how this has been done. This is a crucial point 

Response: Thank you! About how to mapping against the reference genome, we used the HISAT2 and the StringTie.

HISAT2 is a highly efficient alignment system from the RNA sequencing experiment reads, successor to TopHat2/Bowtie2. HISAT uses an indexing scheme based on the Burrows-Wheeler transform and the Ferragina-Manzini (FM) index, utilizing two types of indexes for alignment: a genome-wide FM index to locate each alignment, and many local FM indexes to scale these alignments very quickly, achieving faster speed and less resource usage.

Using StringTie to assemble reads on the contrast pair, StringTie is an algorithm based on optimization theory, using comparison information to construct a multi-variable clipping pattern, using the use of comparison information to construct a traffic network to assemble and evaluate the expression of reads according to the maximum flow algorithm, compared with other software such as Cufflinks, a more complete transcript and better evaluation of expression can be constructed.

After the alignment analysis is completed, the Reads on the StringTie contrast pair are used to assemble and quantify. The analysis process is shown in the following figure

637, data is already a plural noun

Response: Thank you! We have corrected it.

649, confirm if it was necessary a DNAse digestion of the purified RNA used in qPCR to remove residual genomic DNA (it could be present as low contamination even after RNA magnetic beads capture)

Response: Thank you! We are sorry to that we wrote this reverse transcription kit wrong, we have corrected it.We used the PrimeScriptTMRT reagent Kit with gDNA Eraser(TaKaRa, China) to remove residual genomic DNA.

659, which kind of extraction did the samples undergo to ?

Response: Thank you! we have corrected it.

743, check the reference nr. 7 since the authors given names are exposed and not family names

Response: Thank you! We have checked the names and corrected them.

Reviewer 2 Report

The manuscript "Integrated transcriptome and metabolome dissecting interac-2 tion between Vitis vinifera L. and grapevine fabavirus" is an interesting piece of work on a topic, that has been quite neglected up to now. In particular an effort has been here made towards unveiling some of the molecular basis supporting the dialogue V. vinifera and GFabV, which are still to be decoded.

However, I would suggest to the Authors some modifications, in order to improve clarity and an easy reading of the manuscript.

1) The scientific content of references has to be included into the text, avoiding too much sentences beginning with "Someone et al. said/discovere/have shown". This would allow to better follow the overall meaning of a sentence and its main aim.

2) The data are definitely very interesting, but I would suggest to point out the main findings, instead of a simple list of molecules/genes/transcripts.

3) The section on GC MS results has to be improved and discussed.

4) Similarly for the transcriptomic data, in order to give to the reader an authentic and useful "take home message".    

5) Spelling to be revised throughout the manuscript, several mistakes (e.g. Gobal instead of global for at least 2 times)

7) Increased the overall quality of Discussion section, again with the final aim of a clear and effective "take home message".    

Author Response

The manuscript "Integrated transcriptome and metabolome dissecting interaction between Vitis vinifera L. and grapevine fabavirus" is an interesting piece of work on a topic, that has been quite neglected up to now. In particular an effort has been here made towards unveiling some of the molecular basis supporting the dialogue V. vinifera and GFabV, which are still to be decoded.

However, I would suggest to the Authors some modifications, in order to improve clarity and an easy reading of the manuscript.

  • The scientific content of references has to be included into the text, avoiding too much sentences beginning with "Someone et al. said/discovere/have shown". This would allow to better follow the overall meaning of a sentence and its main aim.

Response: Thank you! We have done some revisions about the quotation of references.

  • The data are definitely very interesting, but I would suggest to point out the main findings, instead of a simple list of molecules/genes/transcripts.

Response: Thank you! We have done some corrections in the discussions to show the main findings.

  • The section on GC MS results has to be improved and discussed.

Response: Thank you!We have revised as suggested.

  • Similarly for the transcriptomic data, in order to give to the reader an authentic and useful "take home message". 

Response: Thank you! We have revised as suggested.

  • Spelling to be revised throughout the manuscript, several mistakes (e.g. Gobal instead of global for at least 2 times)

Response: Thank you! We have checked carefully through this manuscript, and have corrected them.

7) Increased the overall quality of Discussion section, again with the final aim of a clear and effective "take home message". 

Response: Thank you! We have revised as suggested.
